# Molecular Mechanism and Clinical Effects of Probiotics in the Management of Cow’s Milk Protein Allergy

**DOI:** 10.3390/ijms24129781

**Published:** 2023-06-06

**Authors:** Ludovica Cela, Giulia Brindisi, Alessandro Gravina, Francesca Pastore, Antonio Semeraro, Ivana Bringheli, Lavinia Marchetti, Rebecca Morelli, Bianca Cinicola, Martina Capponi, Alessandra Gori, Elia Pignataro, Maria Grazia Piccioni, Anna Maria Zicari, Caterina Anania

**Affiliations:** Department of Maternal Infantile and Urological Science, Sapienza University of Rome, 00161 Rome, Italy; ludovica.cela@uniroma1.it (L.C.); giulia.brindisi@uniroma1.it (G.B.); alessandro.gravina@uniroma1.it (A.G.); f.pastore@uniroma1.it (F.P.); antonio.semeraro@uniroma1.it (A.S.); ivana.bringheli@uniroma1.it (I.B.); lavinia.marchetti@uniroma1.it (L.M.); rebecca.morelli@uniroma1.it (R.M.); bianca.cinicola@uniroma1.it (B.C.); martina.capponi@uniroma1.it (M.C.); alessandra.gori85@gmail.com (A.G.); elia.pignataro@uniroma1.it (E.P.); mariagrazia.piccioni@uniroma1.it (M.G.P.); annamaria.zicari@uniroma1.it (A.M.Z.)

**Keywords:** cow’s milk protein allergy (CMPA), probiotics, microbiota, children, pediatric

## Abstract

Cow’s milk protein allergy (CMPA) is the most common food allergy (FA) in infancy, affecting approximately 2% of children under 4 years of age. According to recent studies, the increasing prevalence of FAs can be associated with changes in composition and function of gut microbiota or “dysbiosis”. Gut microbiota regulation, mediated by probiotics, may modulate the systemic inflammatory and immune responses, influencing the development of allergies, with possible clinical benefits. This narrative review collects the actual evidence of probiotics’ efficacy in the management of pediatric CMPA, with a specific focus on the molecular mechanisms of action. Most studies included in this review have shown a beneficial effect of probiotics in CMPA patients, especially in terms of achieving tolerance and improving symptoms.

## 1. Introduction

The effects of probiotics on both human gut microbiota and the immune system are well known, with potential roles in preventing inflammatory gastrointestinal disorders. Indeed, microbiota are necessary for the integrity of the intestinal mucosal barrier, as well as to maintain a highly functioning gut immune system. Recent studies suggest that the increasing prevalence of FAs in the last half century can be associated with changes in the composition and function of gut microbiota or “dysbiosis” [1]. Gut dysbiosis has proinflammatory effects and can lead to several disorders including FAs [2]. Since gastrointestinal (GI) microbiota are important for the development of FAs, GI microbiota modulation could be used as a therapeutic tool for FAs [3]. It has already been established that probiotics have a beneficial effect on atopic dermatitis (AD) [4] and respiratory allergies [5]. By focusing on the molecular patterns, the aim of this narrative review is to collect all the studies that concern the effects of probiotics in the management of cow’s milk protein allergy (CMPA), the most common FA in infanthood.

## 2. Probiotics

Probiotics are live microorganisms with many beneficial effects on human and animal health. According to Havenaar and Huis In’t Veld, this viable mono or mixed culture of bacteria can affect the host beneficially by improving the properties of the microbiota [6]. This indigenous flora is made of 100–1000 microbial species, with a common core, composed predominantly of bacterial species that are essentially stable throughout adulthood. However, they can evolve and transform over a lifetime, depending on a complex set of factors, such as diet, genome, and lifestyle of the host, as well as antibiotic use [7,8]. The interaction between probiotics and gut microflora serves as a supplement to the host environment to provide protection against various enteric pathogens and to increase the relative numbers of “beneficial bacteria” [9,10]. Probiotics can also influence the host’s immune response by modulating the activation of specific gene expression patterns; moreover, they regulate the acute and chronic inflammation in intestinal mucosal tissue, via vascular endothelial growth factor receptor (VEGFR) signaling [11,12].

## 3. Cow’s Milk Protein Allergy (CMPA)

CMPA is a reaction of the immune system to proteins found in cow’s milk [13]. CMPA is the most common FA in infancy: it affects approximately 2% of children under 4 years of age and an even higher percentage of infants. Prevalence of CMPA decreases to less than 1% in children over 6 years of age [2,13,14,15]. Cow’s milk can be divided into two parts: coagulum (curd), which contains 80% of the CMPs, mainly casein (Bos d 8); and lactoserum (whey), which contains 20% of the CMPs, such as α-lactalbumin (Bos d 4) and β-lactoglobulin (Bos d 5) [16]. Other whey proteins include bovine serum albumin (BSA, Bos d 6), lactoferrin (Lf), and immunoglobulin (Bos d 7). Children with allergies are mostly sensitive to multiple cow’s milk proteins, and more than 50% of the individuals with CMPA produce specific IgEs against casein, β-lactoglobulin, and α-lactalbumin [17,18]. Human IgE antibodies are mainly directed to conformational epitopes; therefore, any changes of the proteins’ structure could influence their immunogenic and allergenic properties, potentially inducing both allergies and tolerance [17]. Casein and whey proteins can be ingested by drinking cow’s milk-based formula or through breast milk [13,19,20,21]. CMPA usually presents during the first few months of life, within days or weeks after the introduction of a cow’s milk-based formula into the diet. Symptoms may also occur when exclusively breastfeeding if cow’s milk proteins from the maternal diet are transmitted through breast milk in sufficient quantities [22]. The symptoms can be caused by “immediate” (early) reactions and “delayed” (late) reactions. Immediate reactions occur from minutes up to 2 h after allergen ingestion and are more likely to be immunoglobulin E (IgE)-mediated [23]. They involve the skin, and respiratory and GI tracts [3], including oral pruritus, urticaria, rhinorrhoea or rhinoconjunctivitis, angioedema of oropharynx, eczema, vomiting, and diarrhea [13]. Delayed reactions manifest up to 48 h or even 1 week following ingestion, and they usually involve non-IgE-mediated immune mechanisms [23]. They include food protein-induced enteropathy (FPE), food protein-induced allergic proctocolitis (FPIAP) [24], and food protein-induced enterocolitis syndrome (FPIES) [25]. Non-IgE-mediated manifestations are less common and mostly involve the GI system [26]; the main symptoms are vomiting, regurgitation, diarrhea, rectal bleeding, feeding difficulties, persistent crying, sleep problems, and failure to thrive [13,27]. The diagnosis of CMPA is based on a complete medical history, physical examination, presence of CMP-specific serum IgE, and/or a positive skin prick test (SPT) [23,25]. In non-IgE-mediated allergies, IgE tests are expected to be negative. Atopy patch tests may provide additional information in these cases, but they are not standardized [27,28]; as additionally, endoscopy is not considered as a routinary examination because macroscopic lesions and histological findings are neither sensitive nor specific for CMPA [23,29]. In both IgE- and non-IgE-mediated allergies, the diagnosis needs to be confirmed or excluded by a challenge procedure (ideally blinded) and/or home reintroduction of cow’s milk in the diet [3,23,24,30]. The management of CMPA is based on a strict dietary elimination of CMP from the infant’s diet and from the mother’s diet in breastfed infants, in order to continue breastfeeding, which remains the ideal choice for allergic infants [3,23,27,31,32,33,34]. In formula-fed infants, the elimination diet usually starts with an extensively hydrolyzed formula (eHF) or extensively hydrolyzed casein-based formula (eHCF) [31]. If there is no improvement in symptoms within 2 weeks, an allergic reaction to the remaining peptides in the eHF must be considered, and an amino-acid formula (AAF) should be tried [35,36,37,38,39,40,41,42]. Other possible alternatives are hydrolyzed rice formula (HRF) and soy formula (SF) [3]. Oral immunotherapy (OIT) has been studied as a valid strategy to induce immunological and clinical tolerance to CMP in children with CMPA [43]. OIT consists in the daily administration of the maximum tolerated dosage of the offending allergen, previously established through an OFC. After 6–9 months of OIT, another OFC is performed to test the desensitization, while the achievement of tolerance is assessed through an OFC performed after a variable period of the allergen elimination [44]. OIT apparently involves B-reg and T regulatory cells (Treg), and it leads to a decrease of CMP-specific IgE levels and to an increased level of IgG4 [43,45,46].

### Pathogenesis of FA

Since 2000, the prevalence of allergic and autoimmune diseases has risen in developing countries, now affecting more than one billion people worldwide [47]. Initially, this phenomenon was explained by the “hygiene hypothesis”, which suggests that this higher prevalence of allergic and autoimmune diseases in westernized countries may be associated with the reduced exposure to pathogens and with the decreased incidence of infections due to excessive hygiene measures [47,48]. Recently, the “epithelial barrier hypothesis” was introduced, and it proposes that genetic predisposition, exposure to factors damaging the epithelial barrier, and peri-epithelial chronic inflammation are responsible for the development of allergic diseases [49]. The epithelial barrier acts as a first-line physical defence against pathogens and allergens. Its dysfunction may be caused by many alterations, like changes in lipids and structural proteins, loss of microbiota diversity, or elevated skin pH, as seen in patients with AD. Epithelial barrier derangement can influence its leak, resulting in dysbiosis of microbial content, and translocation of this content into subepithelial and interepithelial compartments, inducing microinflammation [47,48,50].

The normal immune response to ingested food is tolerance. A defect in oral tolerance can lead to FAs [51]. The mechanisms that can lead to the breakdown of oral tolerance are poorly understood. Apparently, multiple pathways are involved. It is now largely demonstrated that the intestinal microbiota have an important role in the functioning of our immune system and that gut dysbiosis, especially if it happens early in life, can lead to the development of inflammatory conditions such as allergies [4]. When food reaches gut lumen, antigens are translocated from the gut lumen into the submucosa via specialized intestinal epithelial cells (IECs), in a process called transcytosis; in lamina propria, mononuclear phagocytes (MNPs), thanks to their dendrites, sample luminal antigens. MNPs transfer luminal antigens to dendritic cells (DCs) that display antigens to naive T cells (CD4+) [52]. They can promote an inflammatory response differentiating in T helper type 1(Th 1), T helper type 2 (Th2), or T helper type 17 (Th17), or they can promote a regulatory response differentiating in Treg. The production of interleukin (IL)-1, IL-6, IL-12, interferon (IFN)-γ, and tumor necrosis factor (TNF)-α promotes Th1 differentiation; IL-17A, IL-17F, IL-21, IL-22, and IFN-γ promote Th17 differentiation; IL-4, IL-5, IL-13, IL-9, and IL-6 promote Th2 differentiation involved in allergic reactions [4,53,54]. In susceptible patients, when allergens reach the epithelial barrier, they stimulate the production of cytokines like thymic stromal lymphopoietin (TSLP), IL-33, and IL-25. These cytokines induce a Th2 differentiation by stimulating group 2 innate lymphoid cells (ILC2) and basophils to produce Th2 cytokines like IL-4, IL-5, IL-9, and IL-13. The activated T cells differentiate into Th2 cells, which are involved in tissue inflammation, and type 2 follicular helper T (Tfh) cells [55]. Tfh cells are a type of T cell localized in B-cell follicles, called follicular helper T (Tfh) cells, which play a critical role in germinal center (GC) reactions [56,57,58,59]. Tfh cells are characterized by the high expression of CXC chemokine receptor 5 (CXCR5), programmed cell death protein 1 (PD-1), B-cell lymphoma 6 (Bcl-6), and IL-21 in both mice and humans. Expression of CXCR5 drives migration in the B-cell follicle by binding CXC motif chemokine ligand 13 (CXCL13), where Tfh cells interact with B cells, regulating antibody isotype switching, affinity maturation, and B-cell memory generation [59,60]. In a mouse model used to test for a peanut allergy, genetic depletion of Tfh cells compromised IgE production and protected mice from anaphylaxis, without affecting Th2 cells [61]. In addition to the effector CD4 T cells, recent evidence demonstrates that Tfh cells are heterogeneous and can be classified into categories [62]. Tfh cells elicited in type 2 responses (Tfh2 cells or type 2 Tfh cells) express higher levels of IL-4 compared to Tfh cells in type 1 responses. IL-4+ Tfh cells are involved in both helminth infections and allergic sensitization, but it was recently found that allergens drive the differentiation of another IL-4+ Tfh cell population, characterized by the co-expression of IL-4 and IL-13 [55,63]. These cells, called Tfh13 cells, are distinct from Th2 and Tfh2 cells, characterized by concomitant high expression of IL-4, IL-5, IL-13, and canonical transcription factors of both classical Tfh cells and Th2 cells, BCL-6 and GATA binding protein 3 (GATA-3), respectively [63]. IL-13 expression in Tfh has been found in mouse models with peanut allergies [64]. Isolated deletion of Tfh13 cells or IL-13 in Tfh cells in Alternaria-sensitized mice modestly affects total or low-affinity IgE levels, but abrogate the production of anaphylactic, high-affinity IgE [63]. Moreover, the canonical Tfh cell cytokine IL-21 seems to work as a negative regulator of IgE class switching [65,66]; indeed, Tfh cells express this cytokine abundantly in type 1 immunizations compared to type 2 responses. Thf13 cells have lower expression of Il-21, which may be an additional mechanism by which these cells promote allergic IgE [63]. The antibody response is also regulated by another population called T follicular regulatory (Tfr) cells, a subset of Treg cells, that reach B cell follicles thanks to their CXCR5 expression [67,68,69]. Tfr cells express the signature transcription factors of both Tfh cells (BCL6) and Treg (Forkhead box P3, FOXP3) [55]. Tfr cells have been shown to inhibit as well as promote IgE response. Tfr cells produce neuritin, a neurotrophic factor, that inhibits phosphorylation of adaptors of IL-4 and IL-13 signaling and limits IgE class switching [70]; on the other hand, the authors of a contrasting report found that IgE production was completely abrogated in the absence of Tfr cells due to the absence of IL-1 [64]. However, further studies are needed to consolidate our understanding of the role of Tfr cells in allergic disease.

Antigen food-specific IgE binds to the FcεRI receptors on mast cells and basophils [Figure 1] [55,71]. Subsequent exposure to food antigen leads to cross-linking of IgE and the IgE receptors on the surface of mast cells and basophils, resulting in the release of preformed mediators into circulation, such as histamine, heparin, and proteases [72]. As discussed before, the presentation of antigens to CD4+ T cells can also lead to a regulatory response. The production of food-antigen specific Treg is the primary mechanism regulating food tolerance [52]. In non-allergic infants, retinoic acid (RA), transforming growth factor β (TGFβ), and IL-10 generate an anti-inflammatory environment, where DCs display food antigens to naive T cells. This condition permits the generation of food antigen-specific Treg, which are fundamental for food tolerance due to downregulation of IgE synthesis. They are also responsible for the unresponsiveness of the immune system to self-peptides. Their proliferation is influenced by IL-10 [4,52]. C-X-C motif chemokine receptor 1 (CXCR1) stimulates the production of IL-10 by MNPs [52]. Treg have a role both in local and in systemic tolerance and produce TGFβ in the gut, which leads to the secretion of IgA by B cells. The generation of food-specific immunoglobulin A (IgA) antibodies may promote oral tolerance by excluding luminal food antigens, providing for mucosal immunity [52,53]. Moreover, some Treg can exit the gut and, through the vascular or lymphatic system, can promote systemic tolerance to food antigens [52].

## 4. Microbiota

Microbiota are defined as the assemblage of living microorganisms that are present in a specific environment [73]; more specifically, they are a community of commensal, symbiotic, and pathogenic microorganisms living in the body other environments [74].

Gut microbiota refer to all the species of microorganisms that colonize the human gut. This colonization starts before birth: the mother transfers microorganisms to the newborn through the placenta, intestine, meconium, and vagina; then, microbiota may evolve in a complex and dynamic way, being influenced by environmental factors [4,75]. They are mostly composed of anaerobic bacteria, and they can be modified by diet and drugs like antibiotics [54]. The dominant founder bacteria of a newborn microbiota are lactobacilli from the mother’s vagina, in the case of a eutocic delivery, while caesarean delivery introduces bacteria from the skin. The microbiota one receives from their mother affects the immune system [52]. The main metabolites produced by gut microbiota are short-chain fatty acids (SCFAs): acetate, butyrate, and propionate [Figure 2] [76].

Microbiota have many beneficial effects on human health including prevention of autoimmune and inflammatory diseases [3]. SCFAs result from the fermentation of insoluble dietary fibers (like non-digestible carbohydrates) by intestinal microorganisms, and they have a role in preventing FAs [1]. In a healthy individual, butyrate is used as a source of energy by colonic epithelial cells for β-oxidation in the mitochondria; they consume oxygen and contribute to maintaining anaerobic conditions in the lumen. Furthermore, butyrate binds peroxisome proliferator-activated receptor gamma (PPAR-γ) in the colonic cells, and it limits the diffusion of oxygen from the cells to the luminal surface, maintaining the anaerobic conditions. In this way, SCFAs repress inducible nitric oxide synthase (iNOS), resulting in the reduction of the nitric oxide (NO) level and nitrate production in the gut. NO and nitrates are specific energy sources used by pathogenic facultative anaerobia to proliferate. Butyrate can also stimulate Treg cells through epigenetic regulation of a Treg transcription factor called FOXP3 [52]. In pathological situations, a low butyrate level is associated with lower oxygen consumption, lower PPAR-γ activity, higher expression of iNOS, and a higher level of NO and nitrates available for pathogens [54]. Several studies showed that GI microbiota in infants are a prognostic factor for FA development, even before the onset of symptoms. Microbiota with low levels of *Bifidobacteria* and with higher levels of proinflammatory metabolites are more frequently observed in infants who will later develop FAs [3]. Moreover, recent evidence shows that allergic sensitization in infanthood may be linked to gut microbiota with reduced capacity to produce SCFAs and, in particular, butyrate [77]. Recent studies tried to explore the molecular mechanisms underlying the role of microbiota in human health. Behind this phenomenon, there is a close interaction between bacteria metabolites like SCFAs and specific receptors of host cells, which can lead to the inhibition or activation of specific signaling pathways [52]. An important factor for food tolerance is the integrity of the intestinal barrier, which is mainly influenced by IL-22: it regulates mucus secretion and microbial peptide production. In this way, IL-22 fortifies the intestinal barrier, reducing the possibility that FAs reach systemic circulation. The impairment of integrity of the intestinal barrier can facilitate the passage of food antigens through it, which can affect oral tolerance of food [75]. In healthy individuals, the microbiota (in particular, *Clostridia* colonization) stimulates type 3 innate lymphoid cells (ILC3) to produce IL-22. It is conceivable that commensal bacteria can influence the integrity of the gut barrier and, consequently, the allergic sensitization [52]. Microbiota seem to regulate allergic effector cell abundance. In 2010, Herbst et al. showed that germ-free allergic mice had a higher number of basophils, infiltrating lymphocytes, and eosinophils accumulated in their airways compared to the control group [78]. In 2013, Cahenzli et al. reported that antibiotic-treated and germ-free mice had higher serum IgE levels and an increased number of mast-cell-surface-bound IgE [79]. Altered microbiota may lead to an increased histamine level, associated with a range of mucosal inflammatory disorders like FAs [80]. In the GI tract, histamine is largely present, and it has an important role in immunoregulation of some GI disorders including FAs. The effects of histamine depend on the expression and activity of its four receptors: histamine receptor 1 (H1R), H2R, H3R, and H4R. The receptor responsible for many of the features associated with the allergic immediate-type hypersensitivity response is H1R; H2R can modulate mast cell degranulation, antibody synthesis, cytokine production, and T-cell polarization; H3R is an autoreceptor in the nervous system; H4R, the most recent receptor to be discovered, shares some properties with the H3R. The levels of histamine and its receptors could be involved in oral tolerance and sensitization of food, although the mechanisms are not well known [80]. Histamine can modulate immune responses thanks to the different expressions of its receptors: H1R is predominant on Th1 cells, and H2R on Th2 cells. Histamine enhances Th1 responses by triggering H1R. Both Th1 and Th2 responses are downregulated by H2R. In mice without H1R, there is a suppression of IFN-γ, a dominant secretion of Th2 cytokines IL-4 and IL-13, and higher levels of IgE, IgG1, IgG2b, and IgG3 compared with mice lacking H2R [81].Any changes in composition and function of our gut microbiota can lead to a pathological condition called “dysbiosis” [1]. Dysbiosis can occur in three ways: loss of microbial diversity, loss of beneficial bacteria, and expansion of opportunistic pathogens. Gut dysbiosis has proinflammatory effects, can contribute to the breaking of oral tolerance [3], and may have a role in FA development [2], including CMPA. Several studies show that the microbiota in infants with CMPA are different from their healthy counterparts [82]. The gut microbiota of children with CMPA compared to healthy children are richer in *Trichocomaceae*, *Ruminococcaceae*, *Bacteroides*, and *Alistipes*, with a decrease in *Bifidobacterium* [83]. In 2019, Feehley et al. compared germ-free mice colonized with gut microbiota from healthy children to germ-free mice colonized with gut microbiota from infants with CMPA, and the first group had higher protection against anaphylactic responses to cow’s milk allergens. In this way, they demonstrated that microbiota are critical for the development of CMPA, and that intervention in the bacterial gut community can be therapeutic [82].

### 4.1. Role of Probiotics in Allergic Disorders

Several studies have tried to investigate the possible role of probiotics in influencing the course of allergic pathologies such as AD, asthma, allergic rhinitis, and FAs [5,84,85,86]. Some studies have demonstrated the beneficial effect that probiotics have on AD; in particular, compared to a placebo, their administration caused a reduction of AD incidence and relieved AD symptoms, especially in children older than 1 year. Beneficial effects of probiotics depend on the dosage and the length of treatment: Jiang et al. showed that the SCORAD (SCORing Atopic Dermatitis) score, which is used to evaluate AD clinical features, improves more in patients treated for at least 8 weeks, compared to patients treated for a shorter period [87]. A meta-analysis published in 2018 revealed that probiotic preparations containing *Lactobacillus* species had a protective effect in infants with moderate-to-severe AD, especially in patients younger than 36 months [88]. Probiotics supplementation causes an alteration of the composition of gut microbiota; these changes play a key role in the decrease in inflammation and in the consequent improvement of AD symptoms. The mechanism responsible for the beneficial effect of probiotics in AD is still unclear, and further prospective studies are needed [84]. In contrast to the optimistic effect on AD, a meta-analysis showed that probiotics do not cause a significant reduction in asthma and wheeze risk. The same study supposed that the lack of effect of probiotics in reducing the risk of asthma/wheeze may have been due to the specific combinations of strains used in these trials or to an insufficient follow-up length [89]. Regarding FAs, such as CMPA, it has been shown that probiotics have beneficial effects, acting in various ways [Figure 3] [85,90].

### 4.2. Microbiota Modulation

Probiotics can modulate gut microbiota composition in different ways, inhibiting pathogenic bacteria growth by competing with them for nutrients [90] and creating a hostile environment for them [91,92]. There are several mechanisms inhibiting bacterial growth, including production of broad spectrum bacteriocins, antimicrobial activity of biosurfactants, and reduction of pH levels thanks to the SCFAs [93].

### 4.3. Enhancement of the Gut Barrier

Probiotics reinforce the physical intestinal barrier. More specifically, they can increase the number of goblet cells and strengthen the mucus layer [94]. Several probiotics are able to increase mucin expression in human intestinal cells [95,96]; e.g., VSL#3 has shown the ability to increase the expression of MUC2, MUC3, and MUC5AC in HT29 cells [97]. Probiotics reinforce the intestinal physical barrier through the phosphorylation of cytoskeletal and tight junctional proteins, promoted by the interaction with actinin and occluding junctions and maintained by the interaction with actin and ZO-1 (Zonula Occludens 1). This phenomenon affects the ability of pathogenic bacteria to bind and invade IECs [98]. Probiotics modulate the expression of tight junction proteins such as occludins, ZO proteins, and cingulins [99].

### 4.4. Immunological Function

Several studies demonstrated the key role of microbiota in the development of gut-associated lymphoid tissue (GALT), showing that, in germ-free mice, there were Peyer patches with incomplete germinal centers, reduced number of intraepithelial lymphocytes, and reduced production of IgA by plasma cells in gut lamina propria [100].

One of the most important properties of probiotics is the ability to bind IECs; several fragments of some probiotics, e.g., *L. casei CRL 431* and *L. paracasei CNCM I-1518*, can be internalized by IECs thanks to their capacity to bind toll-like receptors (TLRs) [101]. Because of this interaction, proinflammatory responses are reduced by regulating NF-kB signaling and reducing TNF-α. Moreover, probiotics modulate anti-inflammatory responses towards IL-10, DC maturation, TGF-β secretion [100], and binding neutrophil receptors Gpr41 and Gpr43 [102].

An important element of the intestinal barrier is represented by serum IgAs, which are a basic element of the humoral adaptive immune system, specifically at mucosal sites. The main producers of these immunoglobulins are plasma cells localized within intestinal lamina propria [103]. Serum IgAs play various roles, such as binding the mucus layer site, where they lead to the immune exclusion of mucosal antigens [104]. In the intestine, IgAs are also responsible for the process better known as “immune exclusion”: it consists in the capacity of these immunoglobulins to attach to commensal and pathogens bacteria, and toxins, blocking an inflammatory response against them [105,106]. It has been shown that probiotics administration can increase mucosal IgA production [100].

Probiotics are able to affect the intestinal immunologic response mediated by T-lymphocytes; these cells play a fundamental role in protecting against pathogenic microorganisms in the GI system and in regulating the responses against food; their activity is influenced by the commensal microbiota [107]. Oral probiotics administration has shown the ability to induce T-lymphocytes cytokines secretion more than commensal bacteria [108]. These cytokines, more specifically INF-γ and IL-12, promote the activation of the Th1 response by increasing IL-1β, IL-6, IL-8, and TNF-α, and the inhibition of the Th2 response, by lowering IL-4 and IL-5 secretion [108,109,110,111].

### 4.5. Reduction of IgE Levels

Probiotics, if given in early life, reduce IgE serum levels and decrease the risk of atopic sensitization [89]. Probiotics prevent allergies, increasing levels of plasmatic C-reactive protein; this finding has been shown in children with eczema and CMPA treated with probiotics [112]. Probiotics have to be viable to induce all of these results; non-viable probiotics do not survive in the intestinal environment and are quickly removed from intestinal lumen [101].

## 5. Discussion

The evidence available in the literature concerning the use of probiotics in the management of CMPA was collected in this review, leading to conclusions that can be useful for clinical practice. The results are encouraging, showing possible benefits in patients with CMPA. Details are discussed below [Table 1].

### 5.1. Retrospective Studies

In 2019, Guest and Fuller published a retrospective analysis comparing the extensively hydrolyzed whey formula (eHWF) and the eHCF + LGG formula in managing cow’s milk allergies in infants. The study population consisted of a randomly selected cohort of 470 cow’s milk allergic infants who were initially fed with eHWF and 470 cow’s milk allergic infants who were initially fed with eHCF + LGG. They showed that eHCF + LGG-fed infants remained on their initial formula longer than eHWF-fed infants (76 vs. 69%; *p* < 0.02) and continued the elimination diet for a significantly shorter time (8.3 ± 6.7 vs. 10.2 ± 8.9 months; *p* = 0.001). Moreover, after 24 months of formula, more eHWF-fed infants were experiencing gastrointestinal symptoms, eczema, and asthma, compared to those fed with eHCF + LGG (7.1 vs. 3.1%; *p* < 0.02). Therefore, the findings from this study indicate that eHCF + LGG not only appears to be more clinically effective than eHWF, but it also may slow down the allergic march seen in children with cow’s milk allergy [121].

In 2021, Sorensen et al. published a retrospective matched cohort study to examine clinical and healthcare data from The Health Improvement Network database, which contained data from 3499 infants (<12 months of age) with confirmed or suspected diagnosis of CMPA, indexed within 5 years. They were divided into two groups: a study group who received AAF supplemented with symbiotic (*n* = 74) and a control group who received only AAF (*n* = 74). The goal was to analyze the clinical effects of the supplementation of the AAF with the probiotic Bifidobacterium breve M16-V and prebiotics (including chicory-derived oligo-fructose and long-chain inulin). The study group, compared to the control group, had statistically significant improvement in clinical symptoms (−37%, *p* < 0.001), infections (−35%, *p* < 0.001), and medication effectiveness (−19%, *p* < 0.001), and significantly less allergic symptoms (32% vs. 61% overall, *p* < 0.001). Infants receiving symbiotics had a significantly higher probability of achieving asymptomatic management without hypoallergenic formula (*p* < 0.001), with a shorter clinical course of symptoms [124].

### 5.2. Non-Randomized Trials

In 2019, Nocerino et al. published a non-randomized trial on 220 subjects, aged 4–6 years, with a history of CMPA in the first year of life. Among these subjects, 110 were treated with eHCF alone, and 110 were treated with eHCF + LGG. They were then compared to the healthy infants control group. Both study groups had evidence of immune tolerance acquisition to cow’s milk for at least 12 months. A full clinical evaluation was performed, and the validated questionnaires on the Rome III diagnostic criteria (QPGS-RIII) were submitted to the parents. The researchers observed that the eHCF cohort had an absolute incidence of functional gastrointestinal disorders (FGIDs) of 0.40 (robust 95% CI, 0.31–0.50), while the eHCF + LGG cohort had an absolute incidence of FGIDs of 0.16 (robust 95% CI, 0.09–0.23). These data correspond to a relative risk difference of 60% (95% CI, 79% to 40%) for eHCF + LGG vs. eHCF (*p* < 0.001). The incidence rate of FGIDs in the healthy cohort was 0.21 (robust 95% CI, 0.12 to 0.29), lower than that seen in the eHCF group and closer to that of the eHCF + LGG group. This study shows that the supplementation with LGG could lower the occurrence of FGIDs in patients with a history of CMPA [120].

In 2013, Berni Canani et al. posted an open non-randomized trial in which 260 infants (<12 months) with CMPA were prospectively evaluated and subdivided into five groups depending on the formula that was already prescribed by a family pediatrician or physician (1 = eHCF; 2 = eHCF + LGG; 3 = HRF; 4 = SF; 5 = AAF). The primary goal was to evaluate the rate of acquisition of tolerance in children with CMPA, which was assessed by a double-blind placebo-controlled food challenge (DBPCFC) performed after 12 months of an exclusion diet with different formulas. The authors observed that the groups receiving eHCF or eHCF + LGG achieved tolerance at 12 months significantly (*p* < 0.05) more frequently (group 1 = 43.6%; group 2 = 78.9%) than the other groups receiving HRF (32.6%), SF (23.6%), and AAF (18.2%). The binary regression analysis coefficient (B) revealed that the rate of acquisition of tolerance at the end of the study was influenced by the mechanism of CMPA (i.e., being lower in subjects with an IgE-mediated mechanism [B 2.05, OR 0.12, 95% CI 0.06–0.26; *p* < 0.001]) and by formula type, increasing with the use of eHCF (B 1.48, OR 4.41, 95% CI 1.44–13.48; *p* = 0.009) and eHCF + LGG (B 3.35, OR 28.62, 95% CI 8.72–93.93; *p* < 0.001). Subgroup analysis showed that in both IgE and non-IgE-mediated CMPA, the addition of LGG resulted in an even higher rate of acquisition of tolerance after 12 months of treatment with eHCF [116].

In 2022, Hubbard et al. published a prospective, single-arm, longitudinal, interventional, multicenter study, conducted between October 2017 and November 2020, in order to evaluate clinical outcomes in infants with CMPA receiving a synbiotic-containing, whey-based eHF (SeHF). The study involved 29 infants (<13 months) with non-IgE-mediated CMPA, currently using or requiring an eHF for the dietary management of CMPA (at least 25% of their energy intake). All infants received a SeHF with galacto-oligosaccharides, fructo-oligosaccharides, and *Bifidobacterium Breve M-16V*, and those who completed the intervention period were evaluated in the follow-up phase of the study during the 6 months before and 6 months after SeHF initiation. They showed significant improvements in GI symptoms, regarding the severity of abdominal pain (in 57%, Z = −2.972, *p* = 0.003), burping (in 46%, Z = −2.321, *p* = 0.02), flatulence (in 79%, Z = −2.802, *p* = 0.005), and constipation (in 14%, Z = −1.890, *p*< 0.04). As secondary outcomes, significant improvements were also observed in rhinitis (41%, *p* = 0.048) and itchy eyes (73%, *p* = 0.048), as well as atopic dermatitis in those infants with severe baseline symptoms. Furthermore, growth and caregiver quality-of-life scores significantly increased (+26.7%, *p* < 0.05) over time. Hospital visits and medications were significantly reduced (−1.61 and −2.23, respectively, *p* < 0.005) in the 6 months after SeHF initiation [125].

In 2023, Strisciuglio conducted a prospective non-randomized pilot trial aiming to investigate the effect of Bifidobacteria in the phenotype and activation status of peripheral basophils and lymphocytes in a pediatric CMPA cohort of 8 children (6–12 months) with a diagnosis of IgE-mediated CMPA. Blood samples were collected at diagnosis (T0), after a 45-day probiotic treatment (T1), and 45 days after the probiotic wash-out (T2). The authors observed that after treating patients with *Bifidobacteria*, naive T lymphocytes decreased significantly. Among the CD3+ cell subsets, both naive and activated CD4+ cells were markedly reduced after taking *Bifidobacteria*, with the lowest percentages at T2. Furthermore, at T1, there was a basophil degranulation in response to all analyzed CMPs, compared to T0. In conclusion, the *Bifidobacteria* treatment was able to modulate both innate and adaptive immunity, with persistent beneficial effects long after the interruption of probiotics oral supplementation. These results suggest that the *Bifidobacteria* could have a role in the acquisition of oral tolerance to CMPs, with a consequent possible benefit in the treatment of CMPA [129].

### 5.3. Randomized Controlled Trials

In 2020, Jing et al. published a randomized double-blind controlled trial in order to assess the possible clinical effect of *Bifidobacterium bifidum TMC3115* in 256 infants from 0.5 to 12 months of age with a diagnosis of CMPA. Half of the patients received *Bifidobacterium bifidum*, and the other half received a placebo. After six months of treatment, in the intervention group, compared with the control group, the allergic symptoms were improved (*p* < 0.05), the serum levels of inflammatory cytokines TNF, IL-1, and IL-6 were reduced, the level of IL-10 was increased (*p* < 0.05), and the IgE serum level decreased (*p* = 0.001). They also studied the composition of the microbiota in both groups: after six months, the probiotic intervention increased the genus proportion of *Lactobacillus*, *Alistipes*, and *Barnesiella*, and reduced the proportion of *Anaerovibrio*, *Christensenellaceae*, *Oscillibacter*, *Bilophila*, *Dorea*, and *Roseburia* when compared with the control group [122].

In 2020, Basturk et al. published a randomized multicenter double-blind placebo-controlled study to show the possible efficacy of LGG in 106 infants (<12 months) with CMPA. They were divided into two groups: a probiotic group (*n* = 51) who received 1 *×* 10^9^ colony forming units (CFU) of LGG orally per day and a placebo group (*n* = 55). After four weeks, infants in the probiotic group showed statistically significant improvement in symptoms like bloody stool, diarrhea, restiveness, and abdominal distension (*p* ≤ 0.001), and improvement in mucous stool (*p* = 0.038) and vomiting (*p* = 0.034). There was no statistically significant improvement in abdominal pain (*p* = 0.325), constipation (*p* = 0.917), and dermatitis (*p* = 0.071). In the placebo group, there were statistically significant improvements in abdominal pain (*p* ≤ 0.001), bloody stool (*p* = 0.007), and restiveness (*p* = 0.026). Complete recovery rates were higher in the probiotic group than in the placebo group, but they were not statistically significant (62% vs. 37%, *p* = 0.147) [123].

In 2017, Berni Canani et al. conducted a parallel-arm randomized controlled trial to test the efficacy of LGG added to eHCF in 220 infants (1–12 months) with IgE-mediated CMPA, randomly divided into two groups: one group received eHCF containing LGG (*n* = 110), and the other one received only eHCF (*n* = 110). After 36 months of LGG supplementation, infants had a significantly lower risk (*p* < 0.001) in developing another allergic manifestation compared to the control group, and a significantly higher probability of acquiring cow’s milk oral tolerance at 36 months (*p* < 0.001). Moreover, they studied the effects of eHCF with LGG on the microbiota: their findings suggested that LGG increased the abundance of butyrate in the microbiota [118].

In 2012, Berni Canani et al. published a randomized trial to study whether supplementation of eHCF with LGG could affect the acquisition of tolerance in infants with CMPA. The subjects of this study were 80 infants (1–12 months of age) evaluated for suspected CMPA but still receiving cow’s milk protein (CMP), randomly divided into group 1 (*n* = 40) receiving eHCF and group 2 (*n* = 40) receiving eHCF + LGG. The study showed that the rate of full clinical tolerance acquisition was always higher in group 2 than in group 1. After 12 months, SPT and atopy patch test (APT) responses were negative in all patients with tolerance acquisition, without significant differences in the two groups. Patients with negative DBPCFC at 6 and 12 months were revaluated to check the persistence of clinical tolerance to CMP, confirming that all subjects consumed regular doses of cow’s milk without signs and symptoms related to CMPA [114].

In 2018, Candy et al. published a multicenter double-blind randomized controlled trial to assess the effects of AAF containing a prebiotic blend (chicory-derived neutral oligofructose and long-chain inulin) and a probiotic strain *Bifidobacterium breve M-16 V* in infants with suspected non-IgE CMPA, compared to an AAF without symbiotic. The aim of the study was to determine if the two different formulas could influence the microbial composition and fecal characteristics (percentages of *Bifidobacteria* and *Eubacterium rectale* (ER)/*Clostridium coccoides* (CC)) of infants with CMPA. They selected 71 subjects (<13 months) with a suspected diagnosis of CMPA, compared to a group of heathy infants. The results of the trial showed that the median percentages of *Bifidobacteria* in the study group were higher (35.4% vs. 9.7%, respectively), and the median percentages of adult-like ER/CC were lower (9.5% vs. 24.2%, respectively) (*p* < 0.001) compared to the control group. Concerning the stool characteristics, the frequency score was lower in the study group than in the control group (*p* = 0.015). Overall, even if GI and general symptoms improved over time, there were not significant differences between the study and control groups. Therefore, the study showed that microbial composition of infants with suspected non-IgE CMPA who received the test formula was closer to that of the healthy infants group than to the control group [119].

In 2013, Vandenplas et al. published a double-blind randomized trial to compare the efficacy of an eHWF enriched with *Bifidobacterium lactis* to eHCF enriched with LGG. The goal of the trial was to show non-inferiority of the eHWF to the eHCF, collect anthropometric data, and analyze gastrointestinal flora, determining concentrations of *Bifidobacteria*, *Lactobacilli*, *Bacteroides*, and *Clostridia*. The participants were 85 infants, randomly assigned to the eHWF formula group (*n* = 41) or the eHCF group (*n* = 44). After one month, an open challenge was performed with a standard starter formula to see if infants showed any reaction. The challenge was positive in 63% of children in the eHWF group and in 75% of children in the eHCF group. Late reactions were observed in 20% of children in the eHWF group and 41% in the eHCF (*p* = 0.037). Moreover, the clinical score used for analyzing symptoms decreased significantly after the first month by 8.07 (95% CI –8.74, 7.40; *p* < 0.001) in both groups. At the age of 1 year, the eHWF group grew faster in terms of weight and height for age z-scores. Therefore, the study showed that the eHWF was as efficacious as the eHCF, and that the administration of these two formulas in infants with mild to moderate CMPA led to a strong improvement in clinical symptoms [115].

In a double-blind randomized clinical trial published in 2014, Ahanchian et al. studied the effects of probiotics in 32 infants (1–12 months old) with CMPA, equally divided into a study and a placebo group. The study group received a symbiotic containing 1 billion CFU of a mixture of *Lactobacillus casei*, *Lactobacillus rhamnosus*, *Streptococcus thermophilus*, *Bifidobacterium breve*, *Lactobacillus acidophilus*, *Bifidobacterium infantis*, and *Lactobacillus bulgaricus*. Before the study began, all patients had GI symptoms. No significant differences were observed when considering daily vomiting or diarrhea between the study and the placebo groups after 72 h (*p* = 0.5 vs. *p* = 0.4), after one week (*p* = 0.5 vs. *p* = 0.4), after two weeks (*p* = 0.6 vs. *p* = 0.3), and after three months (*p* = 0.6 vs. *p* = 0.7). There were no significant differences in rectal bleeding or intestinal colic between the two groups [117].

Hol J. et al. conducted a randomized double-blind placebo-controlled study, which was published in 2008. They enrolled 119 infants (1.4–6 months), divided into two groups: a placebo group that received eHF alone and a probiotic group that received eHF associated with *Lactobacillus casei CRL431* (*Lactobacillus paracasei* subspecies *paracasei*) and *Bifidobacterium lactis Bb-12*. The primary goal of the trial was clinical tolerance, which was obtained in both groups at 6 months (56% of the probiotic group and 54% of the placebo one), showing a non-statistically significant result (*p* = 0.92). Manifestations in other districts were evaluated, showing that the non-responsive infants in the study group expressed symptoms in two or more organ systems (36% skin reactions, 4% subjective reactions, 16% gastrointestinal symptoms). Infants who did not show tolerance at 6 months were rechallenged at 12 months: in the probiotic group, 48% of infants reached tolerance, compared to 60% in the placebo group (non-statistically significant result, *p* = 0.58). The majority of the responsive infants, at 12 months, expressed symptoms in two or more organ systems (27%) or skin reactions (41%); subjective reactions and gastrointestinal symptoms were less frequent (18% and 9%, respectively) [113].

In 2022, Chatchatee et al. published a multicenter, prospective, randomized, double-blind, controlled clinical study to evaluate the possible clinical benefits of AAF supplemented with synbiotics in infants with CMPA. The subjects of this study were 169 infants (<13 months) divided into a SG (*n* = 80), who received AAF + synbiotics (AAF-S) comprising prebiotic oligosaccharides (oligofructose, inulin) and probiotic *Bifidobacterium breve M-16 V*, and a CG (*n* = 89), who received AAF. The study demonstrated that there were no statistically significant differences in the SG and CG in the proportions of subjects who developed tolerance: at 12 months (45% vs. 52%, *p* = 0.401) and at 24 months (64% vs. 42, *p* = 0.530) [126].

In 2023, Nocerino et al. published a randomized controlled trial to evaluate the rate of immune tolerance acquisition in children with CMPA starting dietary treatment with AAF and then switching to EHCF + *Lacticaseibacillus rhamnosus GG* (EHCF + LGG). The subjects of this study included 59 infants (<6 months) with IgE-mediated CMPA, previously placed on AAF by their family pediatrician or physician, divided into two groups: one group who stayed on AAF (*n* = 30) and one group who switched to EHCF + LGG (*n* = 29). After 12 months, the rate of CT was higher in the EHCF + LGG group (0.48, 95% exact CI 0.29–0.67, *n*/N = 14/29) than in the AAF group (0.03, 95% exact CI 0.001–0.17, *n*/N = 1/30). They demonstrated that in IgE-mediated CMPA children, the step-down from AAF to EHCF + LGG was well tolerated and could facilitate the immune tolerance acquisition [127].

In 2023, Yamamoto-Hanada published a double-blind, randomized, two-arm, parallel-group, placebo-controlled phase 2 trial, conducted in Japan to assess the efficacy and feasibility of a combination of heat-killed *Lactiplantibacillus plantarum YIT* 0132 (LP0132) and OIT for treating IgE-mediated CMPA. They enrolled children between 1 and 18 years of age, randomly divided into a study group (*n* = 31), which received citrus juice fermented with LP0132, and a control group (*n* = 30), which received citrus juice without LP0132. Both groups received low-dose slow oral immunotherapy with cow’s milk. After the intervention, 41.4% of the LP0132 group and 37.9% of the control group showed improved tolerance to CM, proven by the CM challenge test at 24 weeks, with no significant differences between the two groups (*p* = 1.00). As secondary outcomes, they studied changes in serum biomarkers: they found significant suppression of sIgG4 reduction in the LP0132 group (*p* = 0.01) and significantly lower levels of IL-5 and IL-9, compared to the control group. Furthermore, focusing on gut microbiota composition, the α-diversity index and Lachnospiraceae increased significantly in the LP0132 group compared to the control group [128].

## 6. Methods

Several studies attempted to demonstrate the role of probiotics in the management of CMPA. This review was written by selecting from the PubMed and Scopus servers the most relevant articles on this topic. The following keywords were used for the research: “cow milk allergy”, “cow milk protein allergy”, “cow’s milk allergy”, “cow’s milk protein allergy”, “CMA”, “CMPA”, and “probiotics”. Studies from 2008 to 2023 were selected, excluding reviews and meta-analyses.

## 7. Conclusions

The possible effects of probiotics in the management of FAs have been investigated in the literature. Most studies included in our review have proven the beneficial effect of probiotics in CMPA patients, especially in terms of achieving tolerance and improving symptoms. Although these preliminary results are encouraging, the differences in study design, time of probiotics administration, and duration of follow-up need to be considered. In conclusion, further studies are needed to confirm the beneficial effects of probiotics in the management of CMPA.

## Figures and Tables

**Figure 1 ijms-24-09781-f001:**
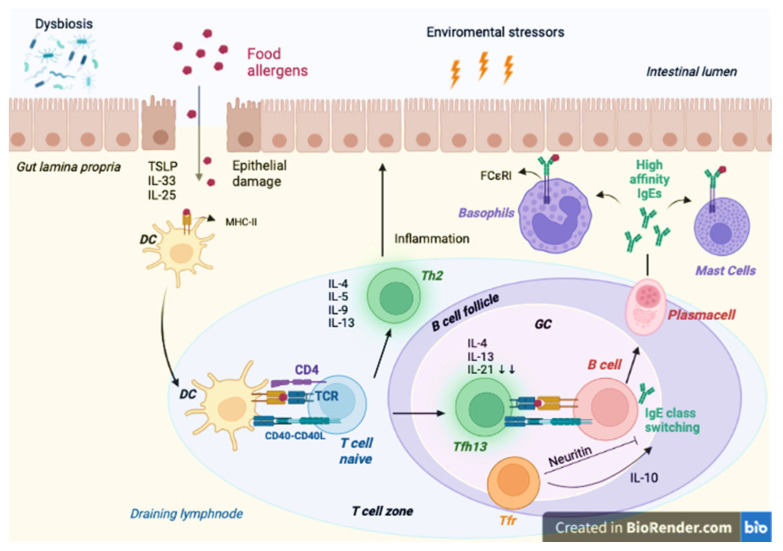
Created with BioRender.com. In susceptible subjects, lack of integrity of the intestinal barrier permits the passage of allergens into sub-epithelial space and the release of proinflammatory cytokines IL-25, IL-33, and TSLP. DCs, under the stimuli of these cytokines, migrate to the draining lymphnode, and present antigens to naive T cells, promoting the type 2 immune response. The activated T cells differentiate into Th2 cells, which are involved in tissue inflammation and interaction with B cells, and into Tfh13 cells, which produce IL-4 and IL-13 in order to elicit B cell isotype class switching to high affinity IgEs. The Tfr cells seem to both suppress and enhance antigen-specific IgEs production by producing neuritin and through IL-10 stimulation, respectively. IgEs bind to the surface of mast cells and basophils through the high affinity IgE receptor Fc epsilon receptor I (FcεR1). The successive exposures to allergens cause the degranulation of sensitized mast cells and basophils, thanks to the binding of IgE to FcεR1 receptors, leading to the release of preformed and de novo synthesized proinflammatory mediators, i.e., leukotrienes, histamine, prostaglandins, and others. CD4: cluster of differentiation 4; CD40: cluster of differentiation 40; CD40L: CD40 ligand; FcεR1: Fc epsilon receptor I; GC: germinal center; IgE: immunoglobulin E; IL: interleukin; MHC-II: major histocompatibility complex class II; TCR: T-cell receptor; Tfh: T follicular helper cells; Tfr: T follicular regulatory cells; Th2: T helper type 2; TSLP: thymic stromal lymphopoietin.

**Figure 2 ijms-24-09781-f002:**
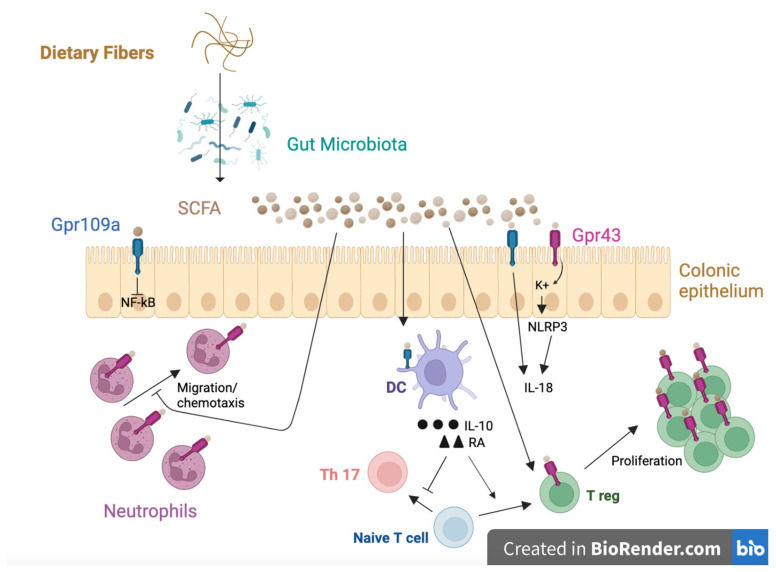
Created with BioRender.com. Dietary fibers are carbohydrates that are not digestible in the small intestines of mammals. Human intestinal lumen is rich in different types of microorganisms, better known as gut microbiota. Dietary fiber fermentation by intestinal microorganisms produces SCFA (acetate, butyrate, propionate). SCFAs bind specific G protein-coupled receptors (GPRs) on the IECs surfaces called Gpr41, Gpr43, and Gpr109a. Recent studies demonstrated the role of these receptors in regulating inflammation, GI functions, allergies, adipogenesis, the central nervous system, and cardiovascular health. Butyrate stimulates DCs to produce IL-10 and RA, which are responsible for the conversion of naive T cells into Treg, suppressing the Th17 response. Treg cells produce IL-10, which has a role in the suppression of intestinal inflammation, and it is required for the induction and maintenance of Treg cells. Gpr43 signaling induces K+ flux, which is responsible for the activation of the NLR family pyrin domain containing 3 (NLRP3) inflammasome, resulting in IL-18 maturation; Gpr109a signaling inhibits nuclear factor kappa-light-chain-enhancer of activated B cells (NF-kB) activation in colonic epithelium and stimulates the IL-18 transcription. Gpr43 activation inhibits neutrophils chemiotaxis, downregulating chemotactic receptors. DC: dendritic cells; Gpr43: G protein-coupled receptor 43; Gpr109a: G protein-coupled receptor 109a; IL: interleukin; NF-kB: nuclear factor kappa-light-chain-enhancer of activated B cells; NLRP3: NLR family pyrin domain containing 3 (NLRP3) inflammasome; SCFA: small chain fatty acids; Th17: T helper type 17; Treg: regulatory T cell.

**Figure 3 ijms-24-09781-f003:**
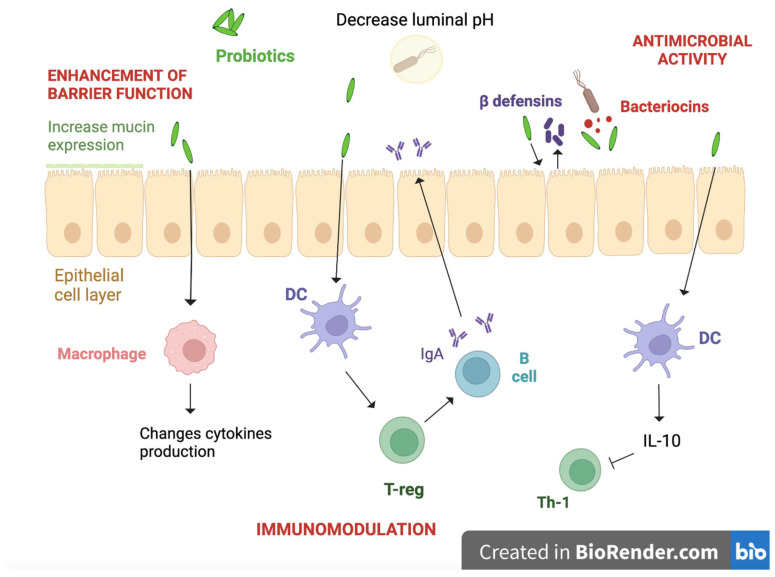
Created with BioRender.com. Probiotics have beneficial effects on FAs through different mechanisms. They increase mucin expression in intestinal cells, and straighten mucus layers and phosphorylate cytoskeletal tight junctional proteins, leading to the enhancement of the barrier function. Immunological functions of probiotics include development of GALT, increased mucosal IgA production, suppression of proinflammatory responses, and activation of anti-inflammatory cytokines like IL-10. Antimicrobial activities of probiotics include decreasing luminal pH and secreting bacteriocins and β-defensins, which are antibacterial substances; these functions lead to the blockage of pathogenic bacteria adherence and translocation. Probiotics compete with pathogenic bacteria for epithelial cell binding sites. DC: dendritic cells; IgA: immunoglobulin A; IL: interleukin; Th-1: T helper type 1; Treg: regulatory T cell.

**Table 1 ijms-24-09781-t001:** Main clinical studies evaluating the effects of supplementation of milk formula with oral probiotics in the management of CMPA in pediatric populations.

Author, Year, Nationality	Study Design	Sample Size at Baseline	Sample Size at Follow-Up	Probiotics	Period of Administration	Follow-Up	Results
Hol et al., 2008, Netherlands [113]	Randomized double-blind placebo-controlled study	119 (1.4–6.0 months) SG = 59 CG = 60	At 6 months: 111 infants At 12 months: 48 infants SG = 23 CG = 25	*Lactobacillus casei CRL431* *Bifidobacterium lactis Bb-12*	4 weeks	6–12 months	At 6 and 12 months, no difference in obtaining CT (*p* = 0.92, *p* = 0.58).
Berni Canani et al., 2012, Italy [114]	Randomized trial	80 infants (1–12 months) CG = 40 SG = 40	Initial groups: CG = 36 SG = 37 At 1 month: CG = 28 SG = 27 At 6 months: CG = 22 SG = 11 At 12 months: CG = 13 SG = 5	*Lactobacillus rhamnosus GG*	12 months	1–6–12 months	Supplementation of eHCF with LGG accelerated CT to CMP.
Vandenplas et al., 2013, Belgium [115]	Double-blind randomized trial	116 infants	85 infants eHWF = 41 eHCF = 44	*Bifidobacterium lactis*, *Lactobacillus rhamnosus GG*	1 month	Until 1 year	eHWF leads to CT faster than eHCF (*p* = 0.037). The SBS decreased significantly (*p* < 0.001) in both groups.
Berni Canani et al., 2013, Italy [116]	Open prospective non-randomized trial	260 infants (0–12 months) eHCF = 55 eHCF + LGG = 71 HRF = 46 SF = 55 AAF = 33	260 infants	*Lactobacillus rhamnosus GG*	12 months	12 months	Achieving CT at 12 months more frequently in eHCF and eHCF + LGG groups (*p* < 0.05).
Ahanchian et al., 2014, Iran [117]	Double-blind randomized trial	45 infants: (1–12 months) SG = 21 CG = 24	32 infants: SG = 16 CG = 16	Symbiotic: *Lactobacillus casei*, *Lactobacillus rhamnosus*, *Streptooccus thermophilus*, *Bifidobacterium breve*, *Lactobacillus acidophilus*, *Bifidobacterium infantis*, *Lactobacillus bulgaricus and FOS*	At least 1 week	72 h, 1 week, 2 weeks, 3 months	No significant differences in daily vomiting or diarrhea in SG and CG (*p* > 0.005). Improvement in rectal bleeding and intestinal colic in both groups (*p* > 0.005).
Berni Canani et al., 2017, Italy [118]	Parallel arm randomized controlled trial	220 infants (1–12 months) SG = 110 CG = 110	At 12 months: SG = 108 CG = 107 At 24 months: SG = 103 CG = 101 At 36 months: 193 infants: SG = 98 CG = 95	*Lactobacillus rhamnosus GG*	36 months	12, 24, 36 months	At 36 months lower risk (*p* < 0.001) in developing another allergy and significantly higher probability of acquiring CT at 36 months (*p* < 0.001).
Candy et al., 2018, UK, Italy, Belgium, Sweden [119]	Multicenter double-blind, randomized controlled trial	122 subjects under 13 months CG = 36 SG = 35 HS = 51	CG = 28 SG = 32	Prebiotic blend of chicory-derived neutral oligofructose and long-chain inulin and a probiotic strain *Bifidobacterium breve M-16 V*	8 weeks	0, 8, 12, and 26 weeks	The stool frequency score was lower in the SG than in the CG (*p* = 0.015).
Nocerino et al., 2019, Italy [120]	Non-randomized trial	330 subjects, (4–6 years) CG = 110 SG = 110 HS = 110	=	*Lactobacillus rhamnosus GG*	12 months	5 years	eHCF + LGG could lower the occurrence of FGIDs in patients with history of CMPA (*p* < 0.001).
Guest et al., 2019, UK [121]	Retrospective cohort analysis	940 infants under 1 year CG = 470 SG = 470	=	*Lactobacillus rhamnosus GG*	Mean duration: 8.3 ± 5.30 months	24 months after starting the formula	eHCF + LGG accelerated CT (*p* = 0.001). Clinical wellbeing in SG infants compared to CG (*p* < 0.02).
Jing et al., 2020, China [122]	Double-blind, randomized controlled trial	256 infants (0.5 to 12 months) SG = 128 CG = 128	At 6 months: 244 infants: SG = 123 CG = 121	*Bifidobacterium bifidum TMC3115*	6 months	6 months	Allergic symptoms improvement in SG (*p* < 0.05).
Basturk et al., 2020, Turkey [123]	Randomized multicenter double-blind placebo-controlled study	106 infants (0–12 months) SG = 51 CG = 55	At 4 weeks: 100 infants: SG = 48 CG = 52	*Lactobacillus rhamnosus GG*	4 weeks	4 weeks	Clinical improvements in both groups
Sorensen et al., 2021, UK [124]	Retrospective matched cohort study	148 infants SG = 74 CG = 74 (0–12 months) in the analyzed database	/	Symbiotic: *Bifidobacterium breve M16-V* and prebiotics (including chicory-derived oligo-fructose and long-chain inulin)	Mean duration: SG: 6.65 ± 5.30 months CG: 8.44 ± 5.62 months	Mean observation period: 1.19 years	In SG, improvement in clinical symptoms and better prognosis (*p* < 0.001).
Hubbard et al., 2022, UK [125]	Prospective single-arm longitudinal interventional multicenter study	33 infants (<13 months)	29 infants	SeHF with galacto-oligosaccharides, fructo-oligosaccharides, and *Bifidobacterium Breve M-16V*	28 days	6 months before and 6 months after SeHF initiation	Improvements in GI symptoms (*p* ≤ 0.005).
Chatchatee et al., 2022, Thailand [126]	Multicenter prospective randomized double-blind controlled clinical study	169 infants (<13 months) SG = 80 CG = 89	At 12 months: 63 infants: SG = 71 CG = 81 At 24 months: 55 infants: SG = 64 CG = 71	Prebiotic oligosaccharides (oligofructose, inulin) and probiotic *Bifidobacterium breve M-16V*	12 months	12–24 months	At 12 and 24 months, no difference in obtaining CT (*p* = 0.401, *p* = 0.53).
Nocerino et al., 2023, Italy [127]	Randomized controlled trial	59 infants (<6 months) AAF = 30 EHCF + LGG = 29	=	*Lacticaseibacillus rhamnosus GG*	Until 12 months of age	12 months	Step-down from AAF to EHCF + LGG could facilitate the CT.
Yamamoto-Hanada et al., 2023, Japan [128]	Double-blind randomized two-arm parallel group placebo-controlled phase 2 trial	61 children (1–18 years) SG = 31 CG = 30	59 children: SG = 30 CG = 29	*Lactiplantibacillus plantarum* YIT 0132 (LP0132)	24 weeks	24 weeks	No significant differences between two groups in CT (*p =* 1.00).
Strisciuglio et al., 2023, Italy [129]	Prospective non-randomized pilot trial	8 infants (6–12 months)	=	*Bifidobacterium Longum BB536*, *Bifidobacterium Infantis M-63*, *Bifidobacterium breve M-16V*	45 days	After 45 days and 45 days after the probiotic wash-out	*Bifidobacteria* could have a role in the acquisition of CT to CMPs.

SG: study group; CG: control group; CT: clinical tolerance; eHCF: extensively hydrolyzed casein-based formula; LGG: *Lactobacillus Rhamnosus GG*; CMP: cow’s milk protein; eHWF: extensively whey hydrolyzed formula; SBS: symptom-based score; HRF: hydrolyzed rice formula; SF: soy formula; AAF: amino acid-based formula; FOS: fructo-oligosaccharides; HS: healthy subjects; FGIDs: functional gastrointestinal disorders; CMPA: cow’s milk protein allergy; SeHF: synbiotic-containing, whey-based eHF.

## Data Availability

Not applicable.

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
