# Peer review of "Molecular Mechanism and Clinical Effects of Probiotics in the Management of Cow’s Milk Protein Allergy"

_ijms, 2023, doi:10.3390/ijms24129781_

Round 1
Reviewer 1 Report
This manuscript discusses how probiotics show promise in managing Cow's Milk Protein Allergy (CMPA) in young children by regulating gut microbiota and modulating inflammatory and immune responses.
Major:
Line 112: IgE binding to Bas/ Mast cells does not induce degranulation!!! Upon subsequent exposure to the allergen, the allergen crosslinks the IgE on the cell surface, causing the cells to degranulate and release preformed mediators such as histamine, heparin, and proteases.
What about T follicular helper cells (Tfh)? They are critical for the development and maturation of B cells.
A paragraph about proteins that cause the allergy should be added, including components (e.g. Bos d 4, 5, 6, 8).
Should be mentioned - “Epithelial barrier hypothesis”, e.g. PMID: 33846604 (Nat Rev Immunol. 2021), PMID: 35108405(Allergy 2022), PMID: 34822880 (J Allergy Clin Immunol. 2022).
This article should mention cow milk (CM)-oral immunotherapy.
Literature is old. Just 14 (out of 89 = 16%) positions are form 2020 or later, and none from 2023. Line 492 – why the literature is selected till 2021? Please add this missing literature from the last two years – this data is so important.
Lines 201-205: Histamine regulates T-cell and antibody responses by differential expression of H1 and H2 receptors gut mucosal immune regulation (Nature 2001 PMID: 11574888, Allergy 2014 PMID: 24286351), and subsequent publications.
Minor:
Figures (all): (1) There is much space on figures, and some text is too small to read when printed in A4 format. Please enlarge. Check the figures in grayscale, e.g. DC (in yellow) is hardly visible. (2) Figures are made with Biorender, and I believe that according to their law, it has to be written down in the publication, that figures were made with Biorender.com. (3) Figure should have all the abbreviations explained. None of the figures has it. (4) The figures should be placed in the page, this way that the whole figure and the legend are on one page, e.g. fig2 is hard to read.
Many abbreviations are unexplained, e.g. IL, Th, IFN-g, IgE. As well as for mistakes, e.g. “Il” instead of “IL” (line 104).
Table 1 has to be better organized, e.g. column “results” is enormous.
Chapter 4.1 The title should be changed. Authors write here mainly about the influence of probiotics on AD, AR, and asthma, but there is almost nothing about CMPA.
Line 222: Authors write about AD, asthma, AR and food allergies, but the citation [5] is just about AR. Please complete.
Line 237-238: Authors write “several studies” that affect Cow’s Milk Protein Allergy (CMPA), but only one article is cited. In addition, the cited article is not about the CMPA, not even once in this article does the word "cow" appear!
Reviewer 2 Report
The authors present a comprehensive review of a very interesting topic. It is clearly written, concise, but offers an extensive overview of the research topic. I`d just suggest they put the Methods section after the Introduction, and rename the Discussion section to perhaps Clinical studies involving probiotics in FA or similar
Title: Molecular mechanism and clinical effects of the probiotics in the management of the Cow’s Milk Protein Allergy, should be: Molecular mechanism and clinical effects of probiotics in the management of Cow’s Milk Protein Allergy.
Just a few clarifications needed:
Abstract: Last studies suggests that the increas-16 ing prevalence of FA can be associated with changes in composition and functions of gut microbiota or “dysbiosis”. should be: Recent studies suggest that the increasing prevalence of FA can be associated with changes in the composition and function of gut microbiota or “dysbiosis”.
Introduction: The effects of probiotics on both human gut microbiota and immune system are well known as well as their role in preventing inflammatory conditions. I wouldn`t say that the role of probiotics in preventing inflammatory conditions is well established. I`d rather formulate this as: and they might have a role in preventing inflammatory disorders, considering that microbiota is necessary for the wellness of intestinal mucosal barrier, and to keep a highly functioning gut immune system.
This seems redundant: Gut microbiota refers to all the species of 30 microorganisms that colonize human gut.
I`d also reformulate this: If gastrointestinal (GI) microbiota is so important for FA development, it is clear that we can use the modulation of the GI micro biota as a therapeutic tool for FA [3]. into: Since gastrointestinal (GI) microbiota is important for the development of FA, GI microbiota modulation could be used as a therapeutic tool for FA [3].
Line 43: Probiotics are bacterial association with beneficial effect on human or animal health. I don`t quite understand what the authors meant, but I guess it should be: Probiotcs are bacteria (or to be exact microorganisms) associated with beneficial effects on human and animal health.
Lines 52-55: this part is unclear: ...........by modulating the activation of specific genetic patterns, and regulate the acute and chronic inflammation in intestinal mucosal tissue, by working on the signalling of vascular endothelial growth factor receptor (VEGFR) [11,12]. Genetic patterns as in expression patterns? (if so, I`d go with gene expression patterns).. and by regulating the acute and chronic inflammation in intestinal....
by working on the signaling of vascular endothelial growth factor receptor (VEGFR).. Working how? If it`s not planned to be discussed in detail, I`d just put "via the VEGFR signaling"...
LInes 77-78: The diagnosis of 77 CMPA is based on a complete medical history, physical examination, presence of CMP-78 specific serum IgE and/or a positive Skin Prick Test (SPT) [20,22] . If non-IgE mediated allergy is discussed, then procedures and tests used for establishing such a diagnosis should be mentioned.
Line 114: As we said before.. I suggest avoiding first person formulation..Instead, something like As discussed before.... could be used.
Line 120: ......down-modulation of IgE synthesis. Do the authors mean down-regulation of IgE synthesis?
The authors are encouraged to have the manuscript reviewed by an English native speaker.
Round 2
Reviewer 1 Report
The authors correctly addressed the comments. The work is much more consistent and valuable.
Thank you.